

# The interactive influence of gender and ergonomic factors, alongside psychosocial associations, on work-related musculoskeletal disorders in Saudi dental students: a cross-sectional study

Saba Kassim[1], Hawazin Mohammad Alblehshi[2,3], Hala Bakeer[1], Manuel Barbosa Almeida[4], Doaa S. Al-Harkan[5], Safa Jambi[1], Doaa Felemban[5], Wafa Alaajam[6,7], Nebras Althagafi[1], Hani T. Fadel[1] and Alla Alsharif[1]

[1] Department of Preventive Dental Sciences, College of Dentistry, Taibah University, Al-Madinah Al-Munawwarah, Saudi Arabia
[2] College of Dentistry, Taibah University, Al-Madinah Al-Munawwarah, Saudi Arabia
[3] Ministry of Health-Alahsa Health Clusters, Alahsa, Saudi Arabia
[4] Egas Moniz Center for Interdisciplinary Research (CiiEM), Egas Moniz School of Health & Science, Almada, Portugal
[5] Department of Oral & Maxillofacial Diagnostic Sciences, College of Dentistry, Taibah University, Al-Madinah Al-Munawwarah, Saudi Arabia
[6] Department of Restorative Dental Sciences, King Khalid University, Abha, Saudi Arabia
[7] Department of Restorative Dental Sciences, Faculty of Dentistry, Sana'a University, Sana'a, Yemen

Corresponding author
Saba Kassim, saba262003@gmail.com

## ABSTRACT

**Background**. Work-related musculoskeletal disorders (WMSDs) among dental students have been documented, with female gender frequently identified as a contributing factor. Yet there is a lack of studies that have investigated the underlying factors between gender and WMSDs.

**Objectives**. The primary aim of this study was to examine the interaction of gender with two specific ergonomic risk factors—academic level and weekly training hours—on WMSDs. Additionally, the associations of psychosocial factors, namely perceived stress and social support, with WMSDs were explored separately.

**Methodology**. A self-reported questionnaire was distributed among a convenience sample of 409 undergraduate dental students at a dental school in Western Saudi Arabia. The questionnaire comprised socio-demographic characteristics, WMSDs using validated questionnaire (Nordic Musculoskeletal Questionnaire), the Perceived Stress Scale, and the Perceived Social Support Scale. Descriptive, bivariate and logistic regression analyses were performed.

**Results**. The median/interquartile range age of the participants was 21 (2) years, and 59% were males. Of the participants, 71% (95% confidence interval (CI) [64.3–76.7]) self-reported WMSDs in at least one area of body over the past 12 months, with the most reported WMSDs being in the lower back, followed by the neck, wrists/hands, and shoulders at 48%, 45%, 31% and 30%, respectively. In fully adjusted logistic regression, being a female and the synergy between gender (female), academic levels
and assigned training hours per week were significantly associated with self-reported WMSDs (adjusted odd ratio (AOR): 0.05, 95% CI [0.02–0.17], $p < .001$; AOR: 1.33, 95% CI [1.07–1.65], $p = .011$).

**Conclusion**. In this study sample, psychosocial factors were not associated with WMSDs. However, the results suggest that female student were more likely to self-report WMSDs than counterparts. Notably, the interaction between gender, academic level and number of hours training assigned per week contributed significantly and positively in self-reported WMSDs specifically among female students. Intervention may consider female students at higher academic levels with training demands.

# INTRODUCTION

Work-related musculoskeletal disorders (WMSDs) comprise a diverse range of inflammatory and degenerative conditions (*de Almeida et al., 2023*). According to the WHO, musculoskeletal disorders affect muscles, ligaments, tendons, joints, nerves, and bones, and are considered work-related when they are caused or aggravated by work environments (*World Health Organization, 2022*). Thus, these disorders are defined as discomfort, disability impairment, or persistent pain in the locomotor system and they come under the umbrella term WMSDs when the work environment promotes their development or aggravation (*Hayes, Cockrell & Smith, 2009*). Among dental students, musculoskeletal disorders have become a prominent concern globally and are referred to as work-related musculoskeletal disorders (*Lietz, Kozak & Nienhaus, 2018*). This is due to performing repetitive hand–wrist motions required for complex dental procedures that demand meticulous precision, often resulting in prolonged physical strain, muscle tension, and poor posture—factors that have been assessed using tools like the Ovako Working posture Assessment System (OWAS) method (*Garcia, Polli & Campos, 2013*) and conceptualized within occupational health criteria, such as those proposed by *Violante (2020)* for diagnosing WMSDs. In addition, dental students often work in constrained spaces, adopting unusual positions that can contribute to poor ergonomics and place immense strain on their musculoskeletal system (*Hayes, Cockrell & Smith, 2009*; *Ng, Hayes & Polster, 2016*). Untreated WMSDs can progress into more severe degenerative and inflammatory conditions that negatively impact daily functioning and quality of life, potentially leading to occupational limitations, increased absenteeism, or even early withdrawal from the dental profession (*Leggat, Kedjarune & Smith, 2007*; *Crawford, Gutierrez & Harber, 2005*).

A systematic review reported the prevalence of musculoskeletal disorders and pain among dentists, dental hygienists and dental students at varying rates between 10.8% and 97.9% in western countries (*Lietz, Kozak & Nienhaus, 2018*). The physical demands of clinical dental work have been linked to a strong association with a high prevalence of

WMSDs among dental professionals, particularly due to repetitive movements, awkward postures, and sustained static positions during patient care (*Hayes, Cockrell & Smith, 2009*; *Lietz, Kozak & Nienhaus, 2018*). In addition, a number of risk factors were associated with WMSDs including age (*Ratzon et al., 2000*; *Alexopoulos, Stathi & Charizani, 2004*), gender, *i.e.,* females were more likely to report WMSDs (*de Almeida et al., 2023*; *dos Santos et al., 2019*), patient procedure treatment time (*Finsen, Christensen & Bakke, 1998*) which may subject students to ergonomic risk factors such as painful postural and repetitive hand or arms movements (*de Almeida et al., 2023*). However, while gender differences in WMSDs might be partly due to biological differences (*de Almeida et al., 2023*), yet, the underlying factors between gender and WMSDs factors namely academic level and hours of training assigned per week has not been investigated.

Evidence from systematic reviews showed that dental students experience stress, which was attributed to the demand of the training and had an impact on the students' well-being (*Alzahem et al., 2011*; *Elani et al., 2014*). *Fava et al. (2019)* and *Lupien et al. (2009)* reported that prolonged exposure to stressful experiences may predispose individuals to accelerated aging and chronic diseases, as a result of cumulative wear and tear on the body that disrupts allostatic load (*McEwen, 2013*). Although these theoretical models are not specific to dental students, they provide valuable context for understanding the potential impact of stress on musculoskeletal health in high-demand professional environments. Notably, in the context of dental students, stress may trigger physiological responses such as muscle tension, which can contribute to the development of WMSDs—a mechanism that has also been observed in broader working populations, as reported in studies examining occupational stress and musculoskeletal outcomes (*Andersen, Haahr & Frost, 2007*; *Hauke et al., 2011*). Likewise, the interplay of stress and WMSDs in dental practitioners has been reported (*Thies et al., 2024*).

Social support is theorized to buffer stressful life events, defined as the assistance received through interactions with others (*Dambi et al., 2018*). Its relevance to WMSDs is underscored by findings in nursing (*Yan et al., 2018*) professionals—a group, like dental students, facing significant physical and psychological demands—suggesting its potential importance in similar healthcare training environments. While some studies have reported an association between psychosocial factors and WMSDs (*Lindfors, Von Thiele & Lundberg, 2006*; *Niu et al., 2023*) others have found no significant link, though they highlight the broader impact of these factors on general health and well-being (*Ylipää et al., 2002*). Studies that link psychosocial factors with WMSDs among dental students are currently scarce.

While global statistics highlight the universal challenges faced by dental professionals, it is important to consider Saudi Arabia's particular context. Research in Saudi Arabia has reported a substantial prevalence of symptoms related to WMSDs among dental professionals, ranging from 54% to 78% (*Alqahtani et al., 2021*; *Zafar & Almosa, 2019*; *Aljanakh et al., 2015*; *Al Wassan et al., 2001*). Saudi Arabia offers a unique context for understanding the prevalence of WMSDs among dental students. Over the years, oral health has been of growing importance, and dental schools have proliferated as a natural consequence.

Increasing emphasis on dental services and a rapidly evolving healthcare landscape have also resulted in more students pursuing dental education in Saudi Arabia (*Al-Shalan, 2018*). This observation, coupled with the inherent challenges of the profession, makes WMSDs an important concern among dental students in Saudi Arabia. Taking part in dental education and completing clinical training have a tangible impact on a student's physical well-being. The limited research available within the Saudi context reveals that WMSDs are not an isolated problem (*Alqahtani et al., 2021*; *Zafar & Almosa, 2019*; *Aljanakh et al., 2015*; *Al Wassan et al., 2001*). Consequently, the Saudi dental education system, while striving for excellence, should also address its students' well-being.

Addressing the prevalence of WMSDs among dental students in Saudi Arabia requires a comprehensive understanding of potential contributing factors. These may include psychosocial elements and the interaction of gender, academic level, and assigned clinical training hours. Investigating these associations can help inform the development of tailored ergonomic intervention programs within dental education. The primary aim of this study was to examine the interaction of gender with two specific ergonomic risk factors—academic level and weekly training hours—on WMSDs. Additionally, the associations of psychosocial factors, namely perceived stress and social support with WMSDs were explored separately.

## MATERIALS AND METHODS

### Study design, sampling, setting and ethical approval

This analytical cross-sectional study recruited a universal convenience sample of all students (203 males and 206 females) at all levels of the Bachelor of Dental Surgery Program at a Dental College in Western Saudi Arabia. In Saudi dental programs, which span six years, students receive laboratory exposure from their first year in courses such as dental anatomy, anatomy, pathology, and microbiology. Clinical experience begins in the fourth year, with students meeting rigorous clinical and laboratory requirements. By the later stages of the program (fifth and sixth year), students are actively involved in patient care. The Research Ethics Committee at the College of Dentistry, Taibah University reviewed and approved the study protocol (TUCDREC/20160204/Alblehshi). The study followed the ethical principles of the World Medical Association Declaration of Helsinki (*World Medical Association, 1964*). An information sheet was given to the students that explained aspects of the study, including its aims, relevance and used methods. The participation of the students was voluntary. Students could withdraw from the study at any time without giving a reason, and without having any impact on their academic achievement. The confidentiality of the information was ensured, and informed consent was obtained from all students before their participation in the study.

### Study questionnaire data collection and participants' recruitment procedures

The data were obtained using a self-administered, paper-and-pencil closed-ended and anonymous questionnaire in English language. The first part of the questionnaire included questions related to respondents' sociodemographic characteristics (*e.g.*, age, gender,

academic level at the dental school, whether one had a re-sit exam, whether one deferred a semester, hours of training in the lab and dental clinic and the use of the left or right hand). The body mass index was self-reported in weight and height (kg/m$^2$) (*Centers for Disease Control, 2024*). The second part asked questions with an explainable picture, to help students understand the appointed areas, of musculoskeletal disorders using the English Nordic Musculoskeletal Questionnaire (NMQ) (*Kuorinka et al., 1987*). The NMQ collected data on self-reported pain in the last 12 months from nine regions of the body *i.e.,* the neck, shoulders, elbows, wrists/hands, upper back, lower back, hips/thighs, knees, and ankles/feet. The validity and reliability and cross-culture validity of the NMQ have been reported previously (*Moodley, Naidoo & Van Wyk, 2018*; *Peros et al., 2011*; *Bao, Winkel & Shahnavaz, 2000*).

Finally, the third part assessed social support and stress using validated questionnaires. Namely the Multidimensional Scale of Perceived Social Support (MSPSS) and the Perceived Stress Scale-10 (PSS-10) (*Zimet et al., 1988*; *Cohen, Kamarck & Mermelstein, 1983*).

The MSPSS was chosen on the basis of being the most widely used tool for assessing social support. It has been extensively translated and adapted across diverse linguistic and socioeconomic contexts, from low-to high-income countries. This tool measures support from three key sources—family, friends, and significant others—and has been widely validated across various cultural settings. While there is limited evidence supporting the psychometric robustness of all its translated versions (*Dambi et al., 2018*), the MSPSS's ease of administration and interpretation ultimately made it an appropriate choice for our study. The MSPSS three domains are: significant other (four items); family support (four items) and friend support (four items). These original domains' items were evaluated on a 7-point Likert scale (1 = very strongly disagree, 2 = mildly disagree, 3 = disagree, 4 = neutral, 5 = mildly agree, 6 = strongly agree, 7 = very strongly agree), with a higher score indicating higher perceived social support. The total score was 84 (range 12–48), with a Cronbach's $\alpha$ of 0.84 (*Zimet et al., 1988*). As for the the PSS-10, it consists of 10 self-reported items that assess situations in an individual's life and are considered stressful. The time frame for this assessment was 'in the last month' and was on a five-point Likert scale (0 = never, 1 = almost never, 2 = sometimes, 3 = fairly often and 4 = very often). The scores ranged from 0 to 40 and were calculated by reversing the scores of four positive items (items 4, 5, 7 and 8). Higher scores represent higher stress levels.

The questionnaire was pilot-tested among 30 undergraduate students in other dental colleges in SA to identify any ambiguities and assess its comprehensiveness. To ensure data quality and motivate student participation, the questionnaire began with information that ensured the anonymity and confidentiality of responses. Additionally, comprehensive and easy-to-understand instructions were provided at the beginning of the questionnaire/survey. Finally, the researchers made themselves available to clarify any questions.

The questionnaires were distributed by hand through a class leader who represented the students of every academic year. The leaders invited the students to participate, and those who agreed were handed the study information sheet and the informed consent form to sign before filling out the questionnaire. However, because participation in this

study was voluntary, students could hand back an incomplete questionnaire or refuse to participate from the beginning. Inclusion into the study was limited to being a registered undergraduate dental student at a dentistry college in one of the universities in Western Saudi Arabia, and willing and agreeing to sign the consent form of the study. Interns at the dental school, pregnant female students and those with musculoskeletal injuries or previous surgery were excluded. In this study, test-retest reliability was not conducted. This was primarily due to the demanding academic schedules of the students at the time of questionnaire distribution, which made it challenging to arrange a follow-up assessment. Test-retest reliability would significantly strengthen future research in this area. The statement Strengthening the Reporting of Observational Studies in Epidemiology (STROBE) was followed (*Vandenbroucke et al., 2007*).

## Statistical analysis

The Statistical Package for Social Sciences version 21 (IBM Corp., Armonk, NY, USA) was used for the data analysis. Descriptive analysis was conducted to report the sample characteristics. Since the continuous variables (*e.g.*, age) did not adhere to normal distribution (Shapiro–Wilk = <0.05) median with an interquartile range (median (IQR)) was reported. The prevalence of complaints per anatomical region in the last 12 month were reported as frequencies and percentages. The bivariate analysis was performed to explore variables associated with the dependent outcome self reported occurrence of WMSDs in the last 12 months (Yes/No) at least one symptom in one of the nine anatomical regions. First, the Mann–Whitney U test was performed to compare the medians of continuous variables. However, when there were ties in medians *i.e.*, the medians values were identical, the mean rank was verified to guide interpretations of the findings (*e.g.*, age Table 1). Second, the chi square test compared the proportions of categorical variables with the dependent variable. Third, variables that were significantly associated with the dependent variable in the bivariate analysis ($p \leq 0.05$) were entered into a multivariable logistic regression to evaluate their association with the dependent variable after adjustment with other variables. However, both the perceived stress and perceived social support as the exposures of interest were forced into regression model irrespective of the *p*-value at the bivariate analysis. As there was multicollinearity between age and academic levels, age was not entered into model, academic levels was entered due to its importance in this study and used as proxy for age. The individual as well as the interaction analysis (Fig. 1) between gender, academic levels and training hours assigned per week for students was performed to test the main and interactive effects of these varaibles on the dependent variable. The continous variable 'hours of training assigned per week' was mean-centered (*Aiken, 1991*). Finally, the results of the logistic regression were reported as odds ratios (OR) with 95% confidence interval (95% CI) and *p*-value $\leq$ .05 was statistically significant.

## RESULTS

### Sample characteristics and distribution with WMSDs

Four hundred and nine students were invited to participate in our study. Of these, 196 declined to participate. The remaining students agreed to participate and proceeded to

**Table 1 Total sample characteristics and bivariate analysis for WMSDs with the sociodemographic and psychosocial factors among dental students ($n = 213$).**

| Categorical variable | Total sample n (%) | WMSDs Yes (n (%)) | P-value[a] |
|---|---|---|---|
| **Gender** | | | |
| Male | 125 (58.7) | 70 (56.0) | **<0.001** |
| Female | 88 (41.3) | 81 (92.0) | |
| **Academic level** | | | |
| Preclinic (Lab only) (years 2,3) | 122 (57.3) | 68 (55.7) | **<0.001** |
| Clinic (Lab and clinic) (years 4,5,6) | 91 (42.7) | 83 (91.2) | |
| **Hand** | | | |
| Right hand | 194 (91.1) | 138 (71.1) | 0.804 |
| Left hand | 19 (8.9) | 13 (68.4) | |
| **Re-sit exam** | | | |
| Yes | 27 (12.7) | 134 (72.0) | 0.332 |
| No | 186 (87.3) | 17 (63) | |
| **Defer a semester** | | | |
| Yes | 199 (93.4) | 142 (71.4) | 0.573[#] |
| No | 14 (6.6) | 9 (64.3) | |

| Continuous variables | Total sample (Median/IQR) | WMSDs | | P-value[b] |
|---|---|---|---|---|
| | | Yes | No | |
| **Age (Median/IQR)** | 21 (2) | 21(2) | 21 (2) | **0.002** |
| **BMI** | 22.27 (5.29) | 22.20 (8.82) | 23.18 (3.68) | 0.275 |
| Training hours assigned weekly | 6 (14) | 18 (14) | 6 (2) | **<0.001** |
| **Perceived stress** | 22 (7) | 22.50 (8) | 21.50 (8) | 0.075 |
| **Perceived social support** | 61 (17) | 61.0 (13) | 62.0 (22) | 0.494 |

Notes.
[a] Chi–square test; Fisher's exact test.
[b] Mann–Whiteny test; values in bold signify $p \leq 0.05$.

data collection, culminating in a final sample of two hundred and thirteen participants, resulting in a response rate of 52%. Male participants were more than females (59% *vs.* 41%). Other characteristics are presented in Table 1. The bivariate findings presented in Table 1 illustrate that female students, older students, seniority within the dental program, *i.e.,* at both clinical and laboratory levels (years 4 and 5 and 6), number of training hours assigned per week were associated significantly with self-reported WMSDs ($p \leq 0.05$). However, both increases in perceived stress and social support were non-significantly associated with WMSDs.

## Distribution of WMSDs

The overall WMSDs self-reported of at least one symptom in one of the nine anatomical regions in the last 12 months by the students were 71% (95% CI [64.3–76.7]). The most
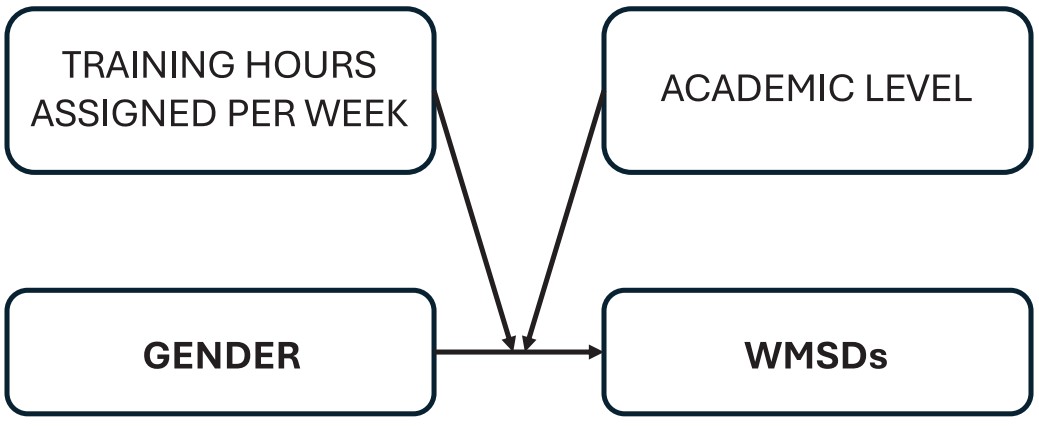

**Figure 1** Conceptual framework of the relationship between gender and WMSDs ($n = 213$).

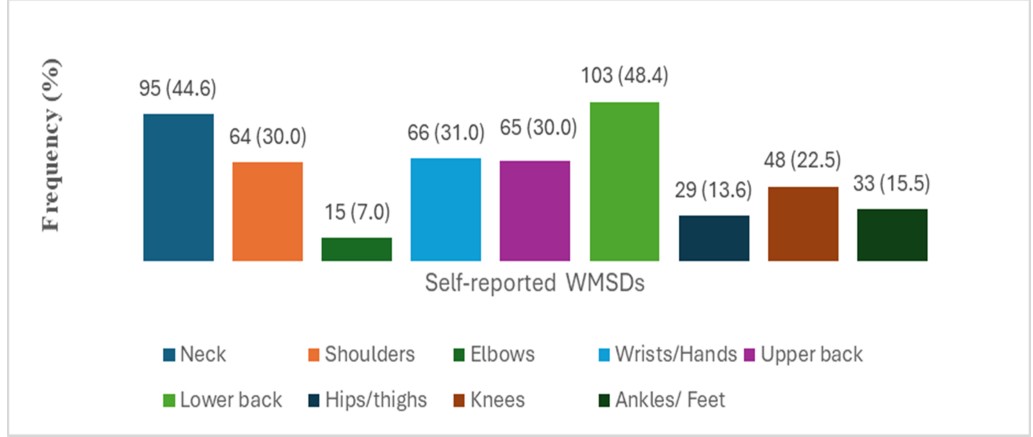

**Figure 2** Frequency and percentage (F(%)) of WMSDs in the last 12 months among dental students ($n = 213$).

distributed WMSDs (Fig. 2) were reported in the lower back and followed by the neck, wrists/hands, and shoulders (48%, 45%, and 31.0%, 30.0%) respectively.

Table 2 shows the results of the binary logistic regression model analysis including the variables, gender, academic level number of training hours assigned per week, perceived stress, perceived social support and the three-way interaction of gender × academic level × number of training hours assigned per week. In the adjusted model gender and the three-way interaction emerged as statistically significantly variables associated with self-reported occurrence of WMSDs in the last 12 months in one of the nine anatomical regions. Male student had significantly lower odds of self-reporting WMSDs compared to female students (AOR: 0.05, 95% CI [0.02–0.17], $p < .001$). In addition, the interaction of being female with increases of academic level and increases in assigned training hours weekly was positively associated with WMSDs (AOR: 1.33, 95% CI [1.07−1.65], $p = .011$).

**Table 2** Binary logistic regression modeling of variables associated with WMSDs among dental students ($n = 213$).

| Variables | B | AOR (95% CI)[a] | p-value |
|---|---|---|---|
| **Gender** | −2.959 | Ref. | **<.001** |
| Male | | 0.05 (0.02–0.17) | |
| female | | | |
| **Academic level** | 0.499 | Ref. | .820 |
| Preclinic (Lab only, years 2,3) | | 1.65 (0.02–120.60) | |
| Clinic (Lab and clinic, years 4,5,6) | | | |
| **Training hours assigned weekly (THAW)** | −.219 | 0.80 (0.59–1.09) | .156 |
| **Perceived social support** | .016 | 1.02 (0.99–1.05) | .274 |
| **Perceived stress** | .006 | 1.01 (0.93–1.09) | .865 |
| **Gender × Academic level × THAW** | .282 | 1.33 (1.07–1.65) | **.011** |

**Notes.**

[a] AOR (95% CI) = adjusted odd ratio with 95% Confidence interval; values in bold signify significant $p < .05$.

The model explained between 30% (Cox and Snell R Square) and 43% (Nagelkerke R Square) of the variance in the WMSDs.

# DISCUSSION

To the best of our knowledge, this study is the first to examine the interactive influence of gender, ergonomic risk factors namely academic level and assigned training hours per week and the association of psychosocial factors, on WMSDs among dental students in Saudi Arabia. Exploration into such aspects was advocated on numerous occasions to further understand WMSDs in a multidimensional manner (*de Almeida et al., 2023*).

## Key findings and ergonomic factors

The current investigation demonstrated that female students, seniority within the dental program, *i.e.,* being at both lab and clinic levels, number of training hours assigned per week were significantly associated with self-reported WMSDs in bivariate analysis. However, logistic regression revealed that only gender and its interaction with academic level and weekly training hours remained significantly associated with WMSDs. This aligns with findings from the systematic review by *de Almeida et al. (2023)*, where gender and training hours were identified as isolated risk factors for WMSDs based on studies included in their review, although a regression analysis could not be performed due to data limitations. Furthermore, these findings are corroborated by *AlSahiem et al. (2023)* among Saudi dental students. One possible explanation could be that females are more conscious and alert about their health than males, making them more inclined and detailed to report changes in their health and well-being. Other explanations are related to inherent differences in muscle tone and energy needs between the genders, making females less resistant to musculoskeletal tension (*de Almeida et al., 2023*).

Our initial expectation was that training hours assigned weekly (THAW) and academic levels would sustain and be a significant predictors of WMSDs; however, this association was not statistically significant in the adjusted regression model. Several reasons might explain this, such as limited variability in reported training hours, potential overlap with

academic level that could have obscured its independent influences, cultural factors and students' adaptation to demanding tasks might lead to underreporting or normalization of discomfort. Despite this, the interaction of gender, academic levels and training hours assigned per week emerged as a significant compound risk of WMSDs instead of individual variables (academic levels and training hours assigned per week). This may highlight that female students reporting WMSDs is conditioned by the impact of increased training hours per week and at higher academic levels. Weekly assigned training hours in relation to WMSDs have been scarcely reported in the current dental students literature (*AlSahiem et al., 2023*). The association of WMSDs with clinical practice has been reported previously (*Hayes, Cockrell & Smith, 2009*; *Al Wassan et al., 2001*; *Chowanadisai et al., 2000*) and has been linked to the physical demands of dental procedures—particularly the precision required, prolonged periods with unsupported arms, and cervical spine rotation and flexion. While earlier studies, such as *Green & Brown (1963)*, described these risks in the context of standing procedures, more recent settings involve seated postures, which may influence the type and distribution of musculoskeletal strain.

## Psychosocial factors and cultural nuances

Even though it was expected that the psychosocial factors might influence WMSDs, our analysis did not found significant associations in this sample. This aligns with a previously reports in a comparable population (*de Almeida, Moleirinho-Alves & Oliveira, 2024*) and might suggest other variables play a major role in WMSDs incidence. Furthermore, while factors like female gender, poor posture habits, inadequate ergonomic knowledge, sedentary lifestyle, high physical activity levels, poor quality of life, and smoking have been described as potential WMSD risk factors, a recent systematic review highlighted that conclusions on their impact remain uncertain due to the low to very low quality of evidence in available studies (*de Almeida et al., 2023*). One should as well considered the cross-culture diversity when using psychosocial scales, *e.g.*, MSPSS (*Dambi et al., 2018*).

Notably, the lack of significant associations between perceived stress, social support, and WMSDs may be partially related to cultural factors in the Saudi context. For instance, a reluctance to express or report stress levels openly, particularly in academic environments, could affect the sensitivity of self-reported measures. Additionally, the generally high levels of perceived social support reported by students may have reduced variability in the data, limiting our ability to detect associations (*Alsubaie et al., 2019*). This challenge in accurately capturing these constructs is further supported by systematic reviews that have recognized the limited psychometric robustness of the MSPSS itself (*Dambi et al., 2018*). Conversely, some studies have shown conflicting results, recovery from lower back pain symptoms was associated with improvement in certain psychological predictors (*George & Beneciuk, 2015*). Such conflicting results point to the necessity for utilization of additional and more specific tools for evaluating psychosocial factors among undergraduate students and their influence on WMSDs.

## Limitations and future research directions

This study included the cross-sectional design, which precluded potential discussions about causality relationships. The self reporting was also associated with possible recall

bias and social desirability. In addition, self-selection into the study may have motivated students' responses. Moreover, the convenience sample, the obtained data from one dental school and the response rate was less than minimum rate (60%), meaning that the study findings cannot be extrapolated to the general undergraduate student population (*Santesso et al., 2020*; *Burns & Kho, 2015*). Additionally, the lack of data on on other factors, such as leisure-time physical activities, may potentially confound the associations between stress, social support, and WMSDs. Future research should address this gap to provide a more comprehensive understanding of these relationships. Notably, different dental procedures demand varying postures and movements which may contribute differently to WMSD development. Future research could therefore specifically focus on this aspect by collecting detailed data on clinical activities and their correlation with WMSD prevalence (*Kumar, Pai & Vineetha, 2020*), and further investigate their interplay with gender, academic levels, and training hours among undergraduate students. Likewise, interventional studies are warranted to empirically evaluate the effectiveness of targeted ergonomic training programs and holistic protocols (*e.g.,* yoga, targeted strengthening exercises) and role of equipment (stool type) and clinical environment in reducing WMSD occurrence (*Verma, Rathore & Yadav, 2023*; *Cherup et al., 2021*; *Danylak, Walsh & Zafar, 2024*) and improving outcomes for dental students. These should also consider the interplay of gender, academic levels, and training hours among undergraduate students. Finally, the primary outcome was the occurrence of at least one WMSD symptom, rather than a specific type of pain (*e.g.,* neck pain). This approach may have obscured the relationship between certain variables and specific body parts. However, the study's strengths stem from investigating the interaction of gender with ergonmic factors that influence WMSDs among female students, as well as the exploring psychosocial factors among dental students using well-established, validated scales in relation to WMSDs.

## CONCLUSION

Within the limits of this study, no significant associations were found between psychosocial factors and WMSDs. However, the findings suggest that the combined influence of gender, academic seniority, and increased clinical training hours may contribute to WMSD risk among female dental students. Further studies are needed to isolate and examine prospectively the specific impact of academic progression and psychosocial stressors on musculoskeletal health in this population.

## ACKNOWLEDGEMENTS

We would like to thank all the undergraduate students who participated in this study. English editing of this manuscript was assisted by Gemini, a large language model by Google.

### Funding

The authors received no funding for this work.

### Competing Interests

The authors declare there are no competing interests.

### Author Contributions

- Saba Kassim conceived and designed the experiments, performed the experiments, analyzed the data, prepared figures and/or tables, authored or reviewed drafts of the article, and approved the final draft.
- Hawazin Mohammad Alblehshi conceived and designed the experiments, performed the experiments, analyzed the data, prepared figures and/or tables, authored or reviewed drafts of the article, and approved the final draft.
- Hala Bakeer conceived and designed the experiments, performed the experiments, authored or reviewed drafts of the article, and approved the final draft.
- Manuel Barbosa Almeida conceived and designed the experiments, performed the experiments, analyzed the data, prepared figures and/or tables, authored or reviewed drafts of the article, and approved the final draft.
- Doaa S. Al-Harkan performed the experiments, authored or reviewed drafts of the article, and approved the final draft.
- Safa Jambi performed the experiments, authored or reviewed drafts of the article, and approved the final draft.
- Doaa Felemban performed the experiments, authored or reviewed drafts of the article, and approved the final draft.
- Wafa Alaajam performed the experiments, authored or reviewed drafts of the article, and approved the final draft.
- Nebras Althagafi performed the experiments, authored or reviewed drafts of the article, and approved the final draft.
- Hani T. Fadel conceived and designed the experiments, performed the experiments, analyzed the data, prepared figures and/or tables, authored or reviewed drafts of the article, and approved the final draft.
- Alla Alsharif conceived and designed the experiments, performed the experiments, analyzed the data, prepared figures and/or tables, authored or reviewed drafts of the article, and approved the final draft.

### Human Ethics

The following information was supplied relating to ethical approvals (*i.e.*, approving body and any reference numbers):

The Research Ethical Committee of College of Dentistry, Taibah University (TUCDREC/20160204/Alblehshi).

## Data Availability

The data is available in the Supplemental File.

## Supplemental Information

Supplemental information for this article can be found online at http://dx.doi.org/10.7717/peerj.19798#supplemental-information.

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
