# Peer review of "The interactive influence of gender and ergonomic factors, alongside psychosocial associations, on work-related musculoskeletal disorders in Saudi dental students: a cross-sectional study"

_PeerJ, doi:10.7717/peerj.19798_

## Round 0.1 · original submission · Major Revisions

Dear authors,

The manuscript presents a relevant investigation into ergonomic risk factors and work-related musculoskeletal disorders (WMSDs) among dental students. While it is well-structured and adheres to ethical and methodological standards, several areas require revision. The introduction and discussion need refinement to enhance clarity and relevance, particularly in linking references to the study context. The research question lacks specificity, and conclusions are not adequately supported by findings. Additionally, deeper statistical analysis, consideration of independent variables, and improved scientific English are recommended. Overall, the study holds merit but requires substantial revision to strengthen its contribution to the field.

·

Basic reporting

1. Used clear unambiguous professional English- very minor grammar, spelling, and expression errors
2. Intro and background literature- a considerable number of references being cited had limited relevance to the context- not relevant to the cohort being researched nor at times to the statement made to that reference.
3. Structure and form acceptable. Lack of highly relevant Figures pertaining to the statistical analysis of the main subject heading. Tables adequate.
4. Self-contained- yes- but with limited results relevant to the hypotheses.
See the annotated pdf for detailed comments.

Basic Reporting -Pass

Experimental design

1. The article conformed with the primary research Aims and Scope.
2. Research question Title lacked clarity to match the stated aims of the project-"to examine the interactive influence of gender, academic level and assigned training hours. Article title stated that "perceived social support and stress" yet did not identify these as 2 specific perceived elements of ergonomic risk. Confusing as to how the research would fill a research gap
Other contextual variables were covered either in limited or no detail at all.
3. Did not feel that a rigorous investigation of the literature had been performed. The research project conformed to the requirements of academic standards both ethically and technically
4. Methods applied were adequate.

Experimental design-pass

Validity of the findings

1. Meaningful replication possible and clearly stated
2. All underlying data was adequate and statistically sound and controlled.
3. Conclusions did not show or prove to have adequate support
4. The conclusion was poorly linked to the original question
4. Did not adequately link the results to the conclusion

Validity of Findings- Fail

Additional comments

1. The Ethics Approval document was not separately submitted- although a reference to the Approval Number was cited in the Questionnaire Approval
2. Deeper analysis and inclusion of other independent variables that may be contributing to WMSD would add possible value- the scope and extent of current ergonomic training, the level of didactic material included dealing with understanding biomechanical dangers; the type of equipment and stools used in the study; the level of teaching and testing competence of indirect vision.

·

Basic reporting

- The manuscript is written in clear, unambiguous, and professional English. Minor typographical and formatting errors (table below shows some examples).

- The introduction provides sufficient context, with a background that frames the research question. The literature is well referenced and relevant, effectively highlighting the gap in understanding WMSDs among dental students in this study context.

- The manuscript conforms to the standard structure following STROBE guidelines, with clear sections for the introduction, methods, results, discussion, and conclusion. Figures (and tables) are relevant, of good quality, and well labelled.

- The manuscript supplies raw data summaries and statistical outputs that appear robust.
* Below is an example of typo errors:
Line Number Error Correction
75 WMSDs 2425 WMSDs 24,25
133 The total score was 84 (range 1-48) I think it is 12 not 1 “The total score was 84 (range 12-48)”
154 did not adhere to normal distribution did not adhere to normal distribution
Table 1 footnote Mann Whiteny Mann-Whitney
212 bivaraite analysis bivariate analysis
221 idividual varaibles individual variables
221 highligt highlight

Experimental design

- This work is original and fits within the scope of this journal. The research question is defined. The study addressed the gap in the literature regarding the interactive effects of gender, academic level, and training hours on WMSDs among dental students.

- The investigation is performed to a good technical and ethical standard. The methods are described in accepted detail, using validated instruments (e.g., NMQ, MSPSS, PSS-10), and allow replication.
Note: The use of a convenience sample from a single institution and a response rate of 52% can be acknowledged as a limitation and discussed regarding their potential impact on generalizability.

- Ethical approval is documented, and the study adheres to established ethical guidelines (e.g., Declaration of Helsinki, STROBE). Informed consent and confidentiality procedures are well described.

Validity of the findings

- The underlying data have been provided and are analyzed using appropriate, statistically sound methods. Descriptive, bivariate, and multivariable analyses (including interaction tests) are presented clearly.

- The manuscript provides sufficient rationale for the research, encouraging meaningful replication. The discussion links the findings to the original research question and appropriately limits conclusions to the supporting results.

- The conclusions are well-stated and linked to the research question. Limitations are candidly acknowledged, including the cross-sectional design, potential self-report bias, sample representativeness, and the lack of data on leisure-time activities. These limitations are appropriately discussed and do not detract from the overall contribution.

Additional comments

The manuscript is well-written and methodologically sound. Its structure, clarity, and comprehensive approach to both experimental design and data analysis make it a valuable contribution to the literature on work-related musculoskeletal disorders among dental students. Only minor language and proofreading corrections are needed.

Reviewer 3 ·

Basic reporting

The manuscript investigates ergonomic risk factors and work-related musculoskeletal disorders among the dental student population. The topic is of interest given the high occurrence of these problems and the methodology appears adequate. However, I recommend performing a deep revision of the introduction and discussion section. In general, the manuscript should undergo a revision of the scientific English, to improve clearness and linearity. I report here some examples of redundancies.

Abstract:
“……dental students and have been repeatedly linked to gender, namely females”: What does this mean?

Please report in the abstract which questionnaires were given (i.e. Nordic Questionnaire).

Introduction lines 46-58. Please simplify the text, which is very repetitive. The abbreviation WMSDs should be given when first mentioned.

References should be improved to include the most relevant articles on the prevalence of WMSDs among dental students and dentists in general. Also, references to support the reasons for WMSD occurrence in selected body areas should be implemented.

Experimental design

Materials and methods:

The sampling section is reported, but no formal analysis is included.

Please report which specialties/treatments the students perform. It would be nice to divide the occurrence of MSDs according to type of procedure performed. If not possible, discuss it in discussion section.

Validity of the findings

Please report which tools were used to consolidate the results (e.g. training of students to answer correctly, motivation, etc.)

Additional comments

Discussion:

I recommend discussing differences in the type of clinical procedure performed (e.g. dental hygiene, restorative, endodontics, surgery). Comment data from literature which reported hypothesis and reasons for higher occurrence of WMSDs and on which body areas. The following articles could support the paper.

Kumar M, Pai KM, Vineetha R. Occupation-related musculoskeletal disorders among dental professionals. Med Pharm Rep. 2020 Oct;93(4):405-409. doi: 10.15386/mpr-1581. Epub 2020 Oct 25. PMID: 33225267; PMCID: PMC7664727.

The use of holistic and integrative protocols (such as yoga intervention) demonstrated to effectively reduce MSDs and to improve flexibility and compliance among dental professionals during different procedures. I recommend preparing a paragraph to report which movements are harmful and which interventions could be useful to reduce the occurrence of MSDs.

Verma, Avichal; Rathore, Vipin; Yadav, Nidheesh. Yoga for proprioception: A systematic review. Yoga Mimamsa 55(2):p 107-113, Jul–Dec 2023. | DOI: 10.4103/ym.ym_37_23

Cherup NP, Strand KL, Lucchi L, Wooten SV, Luca C, Signorile JF. Yoga Meditation Enhances Proprioception and Balance in Individuals Diagnosed With Parkinson's Disease. Percept Mot Skills. 2021 Feb;128(1):304-323. doi: 10.1177/0031512520945085. Epub 2020 Aug 3. PMID: 32746736.

---

## Round 0.2 · accepted · Accept

The authors have satisfactorily addressed all of the reviewers’ comments. The manuscript meets the journal’s standards and is now ready for publication.

·

Basic reporting

The manuscript has been revised after the first round of peer review. The English language is now clear, precise, and professionally written ( I recomend to double check the spacings). It provides enough background and context. Relevant citations support the study’s rationale and aims. Standard academic format with the journal’s requirements were followed. Figures and tables are appropriate, clearly labeled, and of acceptable quality. All results related to the stated hypotheses are included.

Experimental design

This is original work and aligns with the scope of the journal. The research question is clearly defined. The study addresses a gap in the literature concerning the combined effects of gender, academic level, and training hours on WMSDs among dental students.

Validity of the findings

The data are provided and analyzed using valid statistical methods and are clearly presented. The study rationale supports replication. The discussion addresses the research question and limits conclusions to the data. Conclusions are clear and relevant. Limitations are acknowledged without affecting the study’s value.

Additional comments

No more comments